# Molecular Modeling Insights into Metal-Organic Frameworks (MOFs) as a Potential Matrix for Immobilization of Lipase: An In Silico Study

**DOI:** 10.3390/biology12081051

**Published:** 2023-07-26

**Authors:** Prasanna J. Patil, Subodh A. Kamble, Maruti J. Dhanavade, Xin Liang, Chengnan Zhang, Xiuting Li

**Affiliations:** 1Key Laboratory of Geriatric Nutrition and Health, Beijing Technology and Business University, Ministry of Education, Beijing 100048, China; prasanna.j.patil57@gmail.com (P.J.P.); 2150021004@st.btbu.edu.cn (X.L.); 2Key Laboratory of Brewing Microbiome and Enzymatic Molecular Engineering, China General Chamber of Commerce, Beijing 100048, China; 3School of Food and Health, Beijing Technology and Business University, Beijing 100048, China; 4Structural Bioinformatics Unit, Department of Biochemistry, Shivaji University, Kolhapur 416004, MH, India; ksubodh30@gmail.com; 5Department of Microbiology, Bharati Vidyapeeth’s Dr. Patangrao Kadam Mahavidyalaya College, Sangli 416416, MH, India; marutijd@gmail.com; 6Beijing Advanced Innovation Center for Food Nutrition and Human Health, Beijing Technology and Business University, Beijing 100048, China; 7Beijing Association for Science and Technology-Food Nutrition and Safety Professional Think Tank Base, Beijing 100048, China

**Keywords:** *Candida rugosa* lipase, metal-organic frameworks, enzyme immobilization, molecular docking, molecular dynamics simulation

## Abstract

**Simple Summary:**

In recent years, a significant amount of attention has been directed toward MOFs as a potential matrix for the immobilization of enzymes. As per the literature, structures based on metal-organic frameworks (MOFs) offer hydrophobic interaction between lipases and the organic component of the MOF. To ascertain the optimal approach for enhancing the robustness and stability of biocatalysts, it is imperative to gain insights into the molecular interactions between enzymes and ligands. In this regard, the computational methodology proves to be an invaluable asset. A review of the literature reveals numerous reports discussing the interaction between lipase and various ligands. However, there is a lack of available research examining the interaction between lipase and MOF. Hence, it is imperative to comprehend the interplay between *Candida rugosa* lipase (CRL) and the Zeolitic imidazolate framework (ZIF-8) in order to investigate its industrial practicality. This article presents a comprehensive investigation of the utilization of molecular modeling methodologies, specifically molecular docking and molecular dynamics (MD) simulation, for the purpose of studying the interaction between CRL and ZIF-8.

**Abstract:**

CRL is a highly versatile enzyme that finds extensive utility in numerous industries, which is attributed to its selectivity and catalytic efficiency, which have been impeded by the impracticality of its implementation, leading to a loss of native catalytic activity and non-reusability. Enzyme immobilization is a necessary step for enabling its reuse, and it provides methods for regulating the biocatalyst’s functional efficacy in a synthetic setting. MOFs represent a novel category of porous materials possessing distinct superlative features that make MOFs an optimal host matrix for developing enzyme-MOF composites. In this study, we employed molecular modeling approaches, for instance, molecular docking and MD simulation, to explore the interactions between CRL and a specific MOF, ZIF-8. The present study involved conducting secondary structural analysis and homology modeling of CRL, followed by docking ZIF-8 with CRL. The results of the molecular docking analysis indicate that ZIF-8 was situated within the active site pocket of CRL, where it formed hydrogen bonds with Val-81, Phe-87, Ser-91, Asp-231, Thr-132, Lue-297, Phe-296, Phe-344, Thr-347, and Ser-450. The MD simulation analysis revealed that the CRL and ZIF-8 docked complex exhibited stability over the entire simulation period, and all interactions presented in the initial docked complex were maintained throughout the simulation. The findings derived from this investigation could promote comprehension of the molecular mechanisms underlying the interaction between CRL and ZIF-8 as well as the development of immobilized CRL for diverse industrial purposes.

## 1. Introduction

Indisputably, one of the most critical biomacromolecules for life is an enzyme, which is significantly more effective than developed catalysts at catalyzing biological processes that support life [1]. Lipase is the most frequently acknowledged enzyme in diverse fields of industrial biotechnology and microbiology, and it is currently regarded as one of the key participants in numerous industrial processes [2,3]. Lipases are extremely adaptable catalysts in industrial biotechnology because of their advantageous selective features. They are presently used in an array of applications, comprising the synthesis of emulsifiers, biodiesel, cosmetics, pharmaceuticals, flavors, and fragrances, as well as numerous organics and lipophilic antioxidants [1,3,4]. Among their many possible uses, lipases’ primary natural function is to catalyze long-chain triglycerides’ hydrolysis into monoacyl-glycerol. However, there are a number of difficulties with enzymes, such as limited recyclability and denaturation outside of their physiological environments, which raises the costs of operation for their intended use [5]. Enzymes, such as lipases, also suffer from temporary or complete catalytic activity loss when under synthetic conditions. They must be employed in immobilized form for easy retention and recycling to use enzymatic procedures at the industrial level economically. Immobilization has also been linked to improved enzyme heat and shear stability. High mass transfer resistance, a propensity for the by-product glycerol to adsorb onto the support matrix, and a lack of operational stability are the key obstacles to industrializing the enzymatic process [6]. A reasonable support choice with advantageous surface properties and pore diameters can address these issues. Because of this, the creation of novel supports has coincided with the rise in interest in procedures requiring the use of immobilized lipases.

Metal-organic frameworks (MOFs) are highly organized porous crystalline hybrid materials made of a particular metal ion (or metal nodes) and organic linker [7]. To build MOFs, transition metals, actinide elements, alkaline earth metals, and p-block elements are typically utilized as metal ions, and amines, nitrates, carboxylates, sulfonates, and phosphates are regularly used as organic linkers. As newly developed functional materials, MOFs have a number of outstanding characteristics, such as controllable ultrahigh porosity, inherent crystalline nature, large specific surface areas and pore volumes, tunable topological structure, extraordinary multifunctionality, uniform aperture size, exceptional optoelectronic features, plentiful binding interaction sites for a chosen reactant, comparatively high thermal, chemical, and mechanical stability, and heat transformation and storage [8,9,10,11]. They are flexible and competent supports to bind with a range of enzymes having diverse dimensions, morphologies, pensile surface groups, and localized charges thanks to their highly ordered topology with nanometer- to micrometer-level accuracy and the homogenous microenvironment within MOFs. As a result, they have received a lot of attention for scientific studies and real-world applications in various domains, embracing chemical catalysis, biosensing and detection, gas adsorption and separation, and drug loading and delivery [12,13,14,15]. Owing to the defense provided by the highly ordered frameworks of the enzymes, the amalgamation of mesoporous MOFs and enzymes displays boosted stability under extremely harsh conditions, which is conducive to enhancing their catalytic performance in extreme environments such as excessively acidic or basic pHs, high temperatures, organic solvents’ presence, etc. [8,9,10].

By clarifying the specifics of these systems at the molecular level, we can increase our understanding of enzymatic processes. Computational chemistry and bioinformatics have solidified as the primary study fields for comprehending such processes at such scales, thanks to the creation and widespread adoption of faster computers and more effective software. In addition, tools such as molecular docking and molecular dynamics (MD) simulation can make predictions about enantioselectivity, potential catalytic deactivations, or compound affinity, which arise from interactions between an enzyme, substrate, and solvent, more rapidly and with less need for expensive materials [16,17,18]. The molecular process and the intermolecular interactions can both be understood using MD modeling [19]. Computational methods have been useful in the literature for elucidating the link between enzyme structure and function. Having a glance at the literature, the manufacture of enzymatic biolubricants using Lipase Eversa^®^ Transform as a biocatalyst was carried out by Cavalcante et al. using a step-by-step docking and MD strategy [17]. According to a molecular docking investigation, the main types of interactions between the ligands and the amino acid (AA) residues that make up the enzyme’s active site and its surroundings include hydrophobic and van der Waals contacts. These interactions were discovered to be in charge of the lipase-ligand complex’s stability through MD simulations, given the scant number of hydrogen bonds generated throughout the simulations [17]. A little shift in the position of the lipase-ligand complexes with respect to their initial conformations over the manufacturing stages is evidence that the chosen docking poses were satisfactory to reflect the ligand conformations in the region of the enzyme active sites. These findings gave a structural understanding of the interactions between fatty esters used as lubricants and lipase Eversa^®^ [17]. Moreover, the structural relevance of the Neprylysin (NEP) enzyme from the bacterial source *Streptococcus suis* GZ1 was examined by Kamble et al., utilizing a range of bioinformatics techniques [20]. NEP’s hypothesized model from *Streptococcus suis* GZ1 was shown by the results of molecular docking and MD simulation to be able to degrade Aβ peptide in a manner comparable to that of human NEP. Therefore, this investigation’s results may be useful in elucidating the precise chemical process of Aβ peptide breakdown by NEP from *Streptococcus suis* GZ1.

Unfortunately, the literature lacks an in-depth study of the molecular interactions between MOF and lipase. A molecular docking study of porcine pancreatic lipase (PPL) and Zeolitic Imidazolate Framework-90 (ZIF-90) was conducted by the authors of the only research paper that has been published yet reporting the interaction of MOF and lipase [21]. Using molecular docking in the active site and lid domain of the PPL’s X-ray crystal structure, one cage of ZIF-90 (Lig A), six Zn-imidazolate complexes of respective ZIF-90 (Lig B), and one Zn-imidazolate complex of respective ZIF-90 (Lig C) structures were simulated in their study [21]. It was discovered that π-cation interactions, hydrogen bonds, and π-π stacking were the key mechanisms by which all applied structures had an interaction with the active site and lid domain. The findings of the computational investigations agreed with the results of the Circular Dichroism (CD) investigation reported earlier.

We contend that MOF-immobilized enzymes’ next generation must be developed while carefully analyzing how enzymes and carriers interact, as these interactions ultimately determine whether competent and high-performance biocatalysis is feasible. The investigation of the structural characteristics of the immobilized enzyme has posed a persistent and unresolved obstacle in the field. Gaining insight into the enzymatic behavior within a spatially restricted environment would greatly facilitate the interpretation of its inherent properties and enable the customization of the system to exhibit specific functionalities. The elucidation of the correlation between molecular structure and property can be better achieved due to the distinctive and exceptional characteristics of high crystallinity and the homogeneous chemical environment. These factors enable the facilitation of specific interactions between enzymes and MOFs, as well as the establishment of a preferred orientation for the enzyme with minimal interference from the matrix material [22]. Accordingly, we postulate that knowledge of particular enzyme-MOF interactions and the resulting interfacial events is crucial when evaluating understandings for improved, user-designed immobilization techniques. To illustrate our theory, we employed molecular docking and MD simulation approaches to examine the molecular-level events that characterize the interaction between the enzyme *Candida rugosa* lipase (CRL) and ZIF-8 MOF. In order to validate the structure’s quality, secondary structural analysis and homology modeling investigations of CRL were conducted. The molecular docking between the CRL and ZIF-8 was carried out, and then an MD simulation of the docked complex was run. Molecular docking studies were used to determine the ZIF-8’s conformational poses within the CRL’s catalytic region as well as the intermolecular interactions’ amount and nature. The stability of the enzyme-substrate complexes under production-related reactional circumstances was then assessed using MD simulation. To the best of our knowledge, this is the first report demonstrating the detailed molecular interactions between MOF and lipase using in silico analyses, and these molecular modeling studies will serve as the foundation for upcoming industrial research.

## 2. Materials and Method

### 2.1. Structure Retrieval and Preparation

CRL’s crystal structure (PDB ID: 1CRL), having a good resolution (2.06 Å), was retrieved from the Research Collaboratory for Structural Bioinformatics (RCSB) Protein Data Bank (PDB) site, https://www.rcsb.org, accessed on 17 April 2023 [19]. This enzyme is a member of the family of hydrolases (carboxylic esterases) and comprises 534 AAs, wherein AAs Ser209, His449, and Glu341 comprise the key catalytic triad essential for hydrolysis of acylglycerides [23,24]. The associated inhibitors of CRL were removed from the crystal structure (PDB ID: 1CRL), and only enzyme analysis was carried out using UCSF Chimera-1.15 [20,25]. The crystallographic information file (.CIF) file for ZIF-8 was procured from the Cambridge Crystallographic Data Centre’s (CCDC) structural database, with a deposit number of 864309, accessed on 17 April 2023. The file format conversion to .mol was performed utilizing the open-source software OpenBabel version 3.1.1. Subsequently, the inhibitors associated with ZIF-8 were removed from the crystal structure, and solely the analysis of ZIF-8 was conducted through the utilization of UCSF Chimera-1.15 [26].

### 2.2. Secondary Structural Analysis and Homology Modeling Studies of CRL

CRL’s gene and AA sequences were extracted from the NCBI PDB [20]. An essential online server, the “SOPMA” program (Self-Optimized Prediction Method) (https://npsa-pbil.ibcp.fr/cgi-bin/npsa_automat.pl?page=/NPSA/npsa_sopma.html, accessed on 18 April 2023) [20,27], was utilized to analyze the secondary structure of the CRL sequence extracted from the National Center for Biotechnology Information (NCBI). This SOPMA analysis can give insight into the individual AAs involved in building the secondary structure with their positions [28]. The structure of CRL was extracted from RCSB [19]. After the completion of the secondary structure analysis, this extracted structure of CRL was further subjected to various online servers to assess its 3D structure and possible stereochemical quality using different analyses such as Protein Structure Analysis (ProSA) (https://prosa.services.came.sbg.ac.at/prosa.php, accessed on 18 April 2023), PROCHECK, VERIFY-3D [20], and ERRAT (https://saves.mbi.ucla.edu/, accessed on 18 April 2023) [29]. After the structure of the CRL was verified, it was employed for molecular docking analysis with the ZIF-8 ligand. Molecular structures were visualized and analyzed interactively with UCSF Chimera-1.15.

### 2.3. Molecular Docking Analysis of CRL with ZIF-8

The molecular docking was accomplished between CRL and ZIF-8 employing Autodock 4.2.6 software (The Scripps Research Institute, San Diego, CA, USA) [20,21]. The flexible residues of CRL, such as Val-81, Phe-87, Ser-91, Asp-231, Thr-132, Lue-297, Phe-296, Phe-344, Thr-347, and Ser-450, were chosen for the molecular docking study. The 3-D structure of ZIF-8 was acquired from the CCDC structural database. The CRL is composed of two identical chains. The selection process involved the identification of Chain A for the purpose of docking. Correspondingly, the protein targets and ligands were subjected to Kollman and Gastiger charge atom calculations, and non-polar hydrogens were subsequently removed. None of the default parameters were changed. Docked conformations were clustered using a tolerance of 2 Å root-mean-square deviations (RMSD). A grid of dimensions 126 × 126 × 126 was established, featuring a grid spacing of 0.375 Å and centered on the coordinates 68.9, 53.896, and −18.833. The flexible residues located at the active site of CRL were designated for analysis. The utilization of the Lamarckian genetic algorithm (LGA) has been observed in the context of molecular docking research [20,21,30]. The maximum value for the number of energies was established as 2,500,000. A total of 100 runs were conducted, with each independent run producing a maximum of 27,000 genetic algorithm operations. The AutoGrid software was utilized to compute the grid maps of proteins. Subsequently, an examination was conducted on the docked complex with the least amount of energy to identify plausible binding locations. UCSF Chimera-1.15 was employed to visualize the complex for additional analysis and a study on MD simulation.

### 2.4. MD Simulation of a Docked Complex of CRL and ZIF-8

MD simulations are a valuable tool for determining the molecular interactions and the best conformations of protein-ligand complexes [20,24,31]. To better understand the interplay between CRL and ZIF-8, particularly with regard to its interaction with ZIF-8, MD simulation of a docked complex of CRL and ZIF-8 was performed employing the Desmond module(Schrödinger package, version 4.1, D. E. Shaw Research, New York, NY, USA, 2022) [32]. For creating a topology file for the MD simulation of the docked complex, the Optimized Potential for Liquid Simulation-All Atom (OPLS-AA) force field was utilized [33,34]. CRL and ZIF-8 complexes were encapsulated in a water layer of TIP3P solvated system [25,35], and 16 Na^+^ ions were brought in for the purpose of neutralizing the system [20]. In order to minimize the energy used, the steepest descent method was employed, and equilibration was conducted thereafter. The temperature and pressure remained constant during the process. The particle-mesh Ewald (PME) algorithm was utilized while running the simulation [25]. After the equilibration of the system, the docked complex was simulated at 310 K for a duration of 100 ns.

PDBeFOLD was used to analyze structural differences between the MD simulation’s initial and final structures [20,36]. The encoded scoring measures, encompassing ligand and protein RMSD, overall and per residue solvent accessible surface area (SASA), root mean square fluctuations (RMSFs) across the residues, and the radius of gyration (Rg), were used to analyze the molecular trajectories of the protein and its complexes. All dynamic runs were visualized via the VMD (Visual Molecular Dynamics) program version 1.9.3 [24,25]. The MD simulation was visualized and processed using UCSF Chimera version 1.15 and PyMOL (http://PyMOL.sourceforge.net/) software version 1.8.4.0 [24,25].

## 3. Results

### 3.1. Secondary Structural Analysis and Homology Modeling Studies of CRL

Proteins’ secondary structure elucidates key characteristics. The SOPMA tool was used to determine the CRL’s secondary structure. The results generated from the SOPMA analysis conducted on CRL indicate that approximately 26.40% of the residues are involved in forming α-helices, 16.29% of the residues contribute to the creation of extended strand regions, and 51.87% of the residues are engaged in the synthesis of random coils (Figure 1). In contrast, the proportion of AA residues constituting β-turn is only 5.43% (Figure 1). The SOPMA analysis reveals that the primary components of the secondary structure are the helices and random coils. However, CRL’s comparatively low α-helix composition may be attributable to the presence of Pro and Gly, which are known to disrupt helix formation [37]. Based on the SOPMA analysis, it has been verified that the model’s predicted quality can be deemed satisfactory, as the total proportion of regions that form random coils and β-turns is notably low (Figure 1), which pertains to the CRL’s sequence. However, more research is required to verify the involvement of such AAs in controlling AA secondary structure.

Upon analysis of CRL’s extracted 3D structure using PROCHECK [20], it was observed that approximately 86.30% of residues were situated within the most favored regions, while 12.10% of residues were located in additional allowed regions. Additionally, 0.90% of residues were found to belong to generously allowed regions (Figure 2). The structure of CRL exhibits a high degree of quality, as evidenced by the fact that 99.30% of AA residues are located in the favored regions, with only 0.70% of residues found in the outer region (Figure 2) [38]. The model’s quality was also assessed using ProSA-webtool [38,39], a validation tool that provides a Z score. CRL subjected to ProSA analysis was assigned a Z Score of −8.7. The Z Score falls within the acceptable range of X-ray and Nuclear magnetic resonance (NMR) investigations, as depicted in Figure 3A [38]. The results of the ProSA II test indicate that the CRL model exhibits a high level of quality, as evidenced by the maximum number of residues exhibiting negative interaction energy (Figure 3B). Furthermore, for the modeled protein, VERIFY 3D (Figure 4) was utilized to ensure that the atomic model (in 3D) was compatible with the AA sequence (in 1D). The acceptable threshold for residues is a mean 3D-1D score ≥ 0.2 [40]. About 86.33% of the residues here had a mean 3D-1D score ≥ 0.2. In light of these findings, it is clear that the proposed model was consistent with its frequency. The ERRAT plot was employed to evaluate the model and the AA environment. The protein is considered high quality when the estimated error value is lower than the 95% rejection criteria. The greater the score, the better the quality. A score above 50 is typically indicative of a high-quality model [29]. Similar to how good high-resolution structures can generate values of 95% or higher, the typical overall quality factor is around 91% for lower resolutions [29]. In this research, the ERRAT plot for the modeled structure yielded an overall quality factor of 91.8095 (Figure 5). This excellent quality and reliability score suggests that the model is worth investigating further. After validating CRL’s structure using online servers, it was determined that the enzyme possesses good quality and is deemed appropriate for employment in molecular docking alongside ZIF-8.

### 3.2. Molecular Docking Analysis of CRL with ZIF-8

A molecular docking simulation was successfully conducted via Autodock 4.2.6 software for ZIF-8 against the selected protein target CRL. The current investigation involves the examination of the 3D configuration of CRL, utilizing the crystal structure (PDB ID: 1CRL). CRL was also structurally refined (Figure 6). The CRL structure consists of 13 strands and 16 helices. The crystallographic structure of ZIF-8 was procured from the CCDC structural database, further refined (Figure 7), and used for the binding study. It has been observed that ZIF-8 occupies the larger binding pocket located at the interdimer interface of the CRL, as shown in Figure 8. The positioning of ZIF-8 was located at the active site pocket of CRL, having Val-81, Phe-87, Ser-91, Asp-231, Thr-132, Lue-297, Phe-296, Phe-344, Thr-347, and Ser-450 having hydrogen bonding interactions (Figure 8, Table 1).

### 3.3. MD Simulation of a Docked Complex of CRL and ZIF-8

MD simulation is a reliable and precise approach for investigating the conformational changes that occur when a molecule is compelled to conform to a target protein [24]. The knowledge of docked complexes stability, bonding interactions between molecules, and binding position is confirmed using this approach. The conformational stability of the simulated protein/protein-ligand complex is analyzed during the course of the simulation in nanoseconds (ns).

MD simulation generates diverse graphical representations, including RMSD, RMSF, and Rg. The presented graphs serve as indicators of structural stability. The RMSD is employed to evaluate the average shift in the location of a group of atoms in one frame relative to another. If the simulation’s fluctuations have finally stabilized at the end of the simulation around some average thermal structure, the RMSD analysis can show it. All MD simulation trajectories are used to determine this value. Before calculating the RMSD, all protein frames are aligned on the reference frame backbone. RMSD (right Y axis) development of a protein is represented in Figure 9. For small, globular proteins, variations on the scale of 1–3 Å are completely acceptable. A noteworthy structural change in the protein during the MD simulation would be indicated by significantly greater alterations. In our study, the RMSD observed for side-chain atoms and the protein backbone were within a range of 1.6–3.5 Å and 1.1–2.6 Å, respectively. The average RMSD for the protein backbone and the side chain were 1.93 Å and 2.84 Å, respectively. The RMSD graph demonstrated that the ligand-protein complex was stable during MD simulation. During the simulation period, the complex exhibited initial fluctuations of up to 30 ns, followed by a period of stability lasting up to 100 ns.

In addition, RMSF-scoring undulations provide a robust method for estimating macromolecular stability, with lower scores indicating higher stability [24,33]. It helps identify specific modifications in a protein chain. The regions of the protein that show the highest fluctuation are highlighted by peaks in the RMSF in Figure 10. It was found that the N- and C-terminal tails of the protein changed the most throughout time. The protein’s α helices and β strands, for example, are more rigid than the protein’s random coils and turns, which are examples of the protein’s unstructured component. As a result, these strands of α helices and β strands see less variation than the loop areas. The protein RMSF ranges between 0.35–3.74 Å and 0.46–4.61 Å, respectively, for both the backbone and the sides. Whereas for ligand RMSF, a minimum fluctuation of 0.69 Å and a maximum fluctuation of 3.77 Å are observed. The protein structure is confirmed to have fewer fluctuations in the present RMSF graph, indicating that it is stable. During simulation, protein secondary structural elements (SSE) such as α helices and β strands were examined. Figure 11 displays the SSE distribution throughout the protein structure by residue index. Analysis of the SSE that contributes to protein stability revealed that total protein SSE was distributed in 26.20% helices, 11.27% strands, and 37.46% coils, with helices dominating over sheets.

Throughout the MD simulation time, the bonding interactions between the CRL and the ZIF-8 can be examined. These molecular interactions can be divided into various categories, and this information is summarized in Figure 12. Hydrogen bonds (H-bonds), hydrophobic interactions, ionic contacts, and water bridges are the different protein-ligand interactions (or ‘contacts’) types. There are specialized subtypes of each major category of molecular interaction. H-bonds are crucial for the binding and stability of ligands. Since H-bonding characteristics have such a significant impact on ligand specificity, they must be taken into account. Four distinct forms of hydrogen bonds exist between a protein and a ligand: backbone donor, backbone acceptor, side-chain donor, and side-chain acceptor. The current geometric criteria for protein-ligand H-bonds are a distance of 2.5 Å between the donor and acceptor atoms (D—H···A); a donor angle of 120° between the donor-hydrogen-acceptor atoms (D—H···A); and an acceptor angle of 90° between the hydrogen-acceptor-bonded atoms (H···A—X). The crucial H-bonding interactions formed by CRL and ZIF-8 between the residues Glu-66, Thr-132, Ser-209, Ser-450, and Asn-451 indicate possible hydrogen bonding sites in CRL (Table 2, Figure 13A,B). Thr-132, Ser-209, Ser-450, and Asn-451 residues show the formation of H-bond interactions with the backbone of ZIF-8.

Figure 14 depicts, during each trajectory frame, which CRL residues interact with ZIF-8. The scale on the right of the plot indicates that some residues make multiple contacts with the ligand (seen here as a darker orange). The other structural and conformational descriptions of the interaction between CRL and ZIF-8 during MD simulation, such as the intramolecular hydrogen bond (IntraHB) formation, SASA, Rg, molecular surface area (MolSA), and polar surface area (PSA), are shown in Figure 15. All these calculations could be very important to know the structural details of the receptor (CRL) and its possible interactions with ligand (ZIF-8) and showed that the docked complex of receptor and ligand is stable throughout the MD simulation period, and all the interactions that are present in the docked complex are maintained during the whole MD simulation period.

## 4. Discussion

Typically, enzymes tend to occupy the largest pores accessible to them, as this offers them a higher degree of movement. The presence of pores facilitates the unfolding of the enzyme, thereby enabling it to perform its catalytic function effectively [7,41]. Nevertheless, it is plausible that the enzyme possesses the capability to undergo structural rearrangements in order to accommodate itself within smaller pores. The reason behind this phenomenon is that the pores of MOF materials possess functional groups that have the capability to interact with the enzyme, thereby facilitating its immobilization. The role of surface chemistry in MOFs is significant, as enzymes exhibit a higher propensity to bind to surfaces that possess complementary functional groups. The choice of solvent can influence the positioning of the enzyme in the simulation, as certain solvents exhibit a greater capacity for interacting with the enzyme compared to others. The spatial distribution of the enzyme can be influenced by the temperature and pressure conditions within the simulation environment. Under elevated temperatures, there is a greater chance for the enzyme to undergo denaturation, resulting in its unfolding and subsequent occupation of the entire pore. At lower temperatures, there is a greater chance that the enzyme will maintain its native conformation and exhibit an increased tendency to fit within a smaller pore. By utilizing spectroscopic methodologies such as nuclear magnetic resonance (NMR) or fluorescence spectroscopy, it is possible to investigate the positioning and organization of the enzyme within the MOF. This approach offers valuable insights into spatial dispersion, precise location, and the interplay between the enzyme and the framework.

In order to ascertain the exact positioning of the enzyme within the MOF subsequent to an MD simulation, it is imperative to conduct an analysis of the simulation trajectory. This analysis entails evaluating the enzyme’s interactions, binding sites, and spatial distribution within the MOF. Various methodologies, including clustering analysis, radial distribution functions, and surface tethering or PEGylation, can be utilized to ascertain the primary positions or configurations of the enzyme within the MOF and to screen potential attachment sites [42]. Furthermore, the determination of the enzyme’s location in molecular simulations is frequently achieved through the process of energy minimization. This implies that the simulation is executed until the enzyme and the MOF attain a state characterized by the lowest possible energy. In this state, the enzyme will be situated in a configuration that maximizes its stability through optimal interaction with the MOF. During MD simulation, the enzyme’s interaction with the MOF can vary based on the presence of binding sites, the characteristics of the enzyme-MOF interactions, and the parameters set for the simulation. The enzyme may be situated in close proximity to the surface of the MOF, within its pores or channels, or the specific active sites or binding pockets if they are present within the MOF structure [15]. The investigation conducted by Wu et al. employed molecular simulations to examine the spatial distribution of glucose oxidase (GOx) within ZIF-8 [43]. The study’s findings indicate that GOx was situated within the larger pores of the ZIF-8. The enzyme molecules were found to be present in the GOx-ZIF-8 nanocomposite, specifically in regions with a size >8 nm. This observation was further validated by utilizing Fast Fourier Transform (FFT) analysis.

A few papers involved crystallization during the immobilization of enzymes with MOFs [43,44,45]; this suggests that MOFs may restrict the enzyme significantly. The following approaches can be taken into consideration in simulation and experimental investigations to address the problem:
(1)One strategy for dealing with this problem in simulation is to employ a coarse-grained enzyme model. By doing so, the enzyme’s size will decrease, and it will be simpler to fit through the pores of the ZIF-8 framework [46,47]. Adding constraints or positioning constraints to the enzyme during the simulation will also replicate the experimental restraint imposed by the framework. In order to limit the mobility of the enzyme within the MOF, this can entail constraining specific regions or atoms of the enzyme. The investigation of the enzyme-MOF system can be facilitated through the development of a comprehensive model that enables the examination of the temporal dynamics and interactions of the enzyme. The enzyme’s confinement within the MOF structure can be effectively replicated by employing external restraints or constraints, such as harmonic potentials or position restraints, which are defined by suitable force field parameters. Measurements such as cavity occupancy, residence times, and others can be used to determine how much confinement the enzyme faces while housed in the MOF. This will shed light on how much restriction and confinement the framework imposes on the enzyme.(2)The following strategies should be taken into account in experiments:
(a)Enzymatic assays, kinetic studies, or other appropriate techniques can be used to examine the stability and activity of the enzyme-MOF system. This will make it easier to evaluate how the MOF framework affects how the enzyme works and how active it is.(b)Modifying MOF characteristics:
(bi)Pore size modification: MOF’s pore size is too small for the enzyme; we can use different MOFs with larger pores that still have a high surface area and stability.(bii)Functionalization: Changing the surface functional groups of the MOF to enhance interactions with the enzyme. Enzyme-MOF interactions may be improved, and dramatic constraints may be lessened.(c)Examining alternative MOF synthesis techniques: To create a MOF structure with bigger pores or more accessible areas for the enzyme in the experimental in-situ encapsulation, different crystallization techniques might be investigated.(d)Time-resolved methods: Using time-resolved methods, such as in-situ spectroscopy or time-resolved X-ray diffraction, to track the enzyme immobilization procedure in real time. In doing so, we can see how the MOF confines and controls the enzyme as it crystallizes.(e)Techniques for characterizing: Using several characterization methods to examine the shape and distribution of the enzyme within the MOF, such as scanning electron microscopy (SEM) or transmission electron microscopy (TEM). This can assist in determining how efficiently the enzyme is restricted and immobilized.(f)Catalytic activity tests: Calculating the immobilized enzyme’s catalytic activity and contrasting it with the free enzyme’s activity. This reveals the MOF’s efficiency in inhibiting the enzyme’s mobility and functionality.

Our understanding of the behavior of enzymes encapsulated within MOFs, as well as the molecular interactions between enzymes and the impact of the support on the enzyme’s structural stability, remains limited. Once the enzyme is introduced into the MOF, establishing confidence becomes challenging, and only a limited number of studies have endeavored to explore this intricate interaction at the molecular scale. The review article by Wang and Liao in 2021 gave some overview regarding this issue [48]. Even though a few studies, including the Mohammad Latif group, found that larger enzymes usually show higher stability in MOFs under extreme conditions, the underlying molecular cause of this situation remains to be explored [49]. The literature survey shows a huge research gap in the molecular modeling studies of MOF-enzyme interaction, and to date, only a few papers published reporting the interaction of MOF with enzymes are available. Perhaps, as we mentioned earlier, ours is the very first study of MD simulation analysis to understand the molecular mechanism underlying the interaction between lipase and MOF. Upon conducting extensive research, a number of questions have emerged, such as: Why is there a scarcity of research attempts in modeling enzyme-MOFs using MD techniques? Additionally, what are the challenges associated with this particular area? The utilization of MD simulations to model enzymes encapsulated within MOFs poses various challenges, potentially accounting for the scarcity of research conducted in this domain. The challenges stem from the complex characteristics of enzymes and MOFs and their interplay. We conducted a thorough literature review to look for answers and came across some noteworthy explanations, which we’ve included below:Size and complexity: Enzymes are characterized by their substantial molecular size, composed of numerous atoms. In contrast, MOFs exhibit a wide range of structural complexity, encompassing varying quantities of metal ions and organic ligands, typically ranging from hundreds to thousands. Simulating large systems with atomistic precision necessitates substantial computational resources and time. The computational costs experience exponential growth as the size of the system increases, thereby imposing constraints on the feasible timescales and the number of replicates that can be simulated.Force field parametrization: The precise determination of force fields for enzymes and MOFs is of utmost importance to ensure simulation reliability. Nevertheless, parameterizing force fields for biomolecules and MOF components presents significant challenges. Enzymes frequently necessitate specialized force fields due to their distinctive functional groups and catalytic mechanisms. In a similar vein, MOFs exhibit a wide range of metal-ligand coordination environments, thereby requiring precise parameterization in order to characterize their properties effectively.Differences in binding free energy: Due to the small difference in free energy between substrate enantiomers, predicting an enzyme’s enantioselectivity towards these compounds is difficult. One probable explanation is that enzymes and MOF have different binding free energies, which are commonly used to quantify the affinity of biomolecular interactions [50].Metal-enzyme interactions: Metal ions are frequently necessary as cofactors for the catalytic activity of enzymes. Accurately representing metal-enzyme interactions in MD simulations poses a challenge, primarily stemming from the requirement for advanced force fields capable of accurately describing the intricate nature of metal coordination chemistry. The process of parameterizing metal centers can present significant challenges, particularly when dealing with transition metal ions that possess intricate electronic configurations.Enzyme flexibility: Enzymes are highly dynamic biomolecules that undergo conformational modifications to execute their essential biological functions. Effectively simulating these conformational changes poses significant computational challenges. The comprehensive exploration of an enzyme’s conformational space within a MOF is frequently difficult due to the restricted timescales that can be accessed through MD simulations.Treatment of solvents and ions: MOF systems are commonly subjected to simulation in the presence of solvents and ions in order to replicate the conditions observed in experimental settings. Precisely representing the solvent environment, encompassing water molecules or organic solvents, is imperative; however, it introduces complexities to the simulation. Furthermore, the selection of ion parameters and concentrations can impact the stability and dynamics of the enzyme-MOF system.Enzyme-MOF interface: The interaction between the enzyme and MOF can exhibit complexity, encompassing various types of interactions, including hydrogen bonding, hydrophobic contacts, and electrostatic interactions. The task of capturing the complex intricacies of these interactions and comprehending the impact of the MOF on enzyme dynamics can present considerable difficulties.System stability and timescales: Enzyme-MOF systems possess inherent instability owing to the possibility of structural rearrangements and destruction of the MOF framework upon binding with enzymes. The challenge lies in simulating the long-term stability of the enzyme-MOF system while ensuring an accurate representation of the system. Numerous enzymatic reactions take place within time frames that surpass the capabilities of conventional MD simulations. The investigation of rare occurrences, such as substrate binding or enzymatic reactions, necessitates the utilization of sophisticated sampling methodologies, such as enhanced sampling or hybrid techniques that integrate MD with other complementary strategies.Limited experimental data: The experimental characterization of enzyme-MOF systems presents significant challenges, resulting in the limited availability of benchmarking data to validate simulation outcomes. The absence of empirical data poses a significant obstacle to the advancement and verification of precise simulation models.

Notwithstanding these obstacles, researchers are actively engaged in overcoming them and propelling the domain of enzyme-MOF MD modeling forward. Current research efforts are focused on investigating novel force fields, refining sampling techniques, and devising hybrid methods that integrate various simulation approaches. These endeavors aim to overcome existing limitations and facilitate more extensive investigations into enzyme-MOF systems.

## 5. Conclusions

The present investigation centered on the employment of computer-aided molecular design and MD simulation methodologies to conduct molecular modeling of CRL with ZIF-8. The study offers a molecular and structural framework that outlines the potential for interaction between CRL and ZIF-8. In this study, we successfully docked the ligand ZIF-8 with the CRL enzyme. The molecular docking study was followed by an MD simulation investigation to assess the structural stability of CRL with the ZIF-8 complex. These molecular modeling studies revealed that the docked complex of CRL with ZIF-8 is stable and possesses good intermolecular bonding interactions throughout the simulation. During our investigation, we first reported the precise structural features and insights into their intermolecular interactions crucial to targeting ligand ZIF-8 and CRL residues such as Glu-66, Thr-132, Ser-209, Ser-450, and Asn-451. We also report the significance of the key conformational changes in the docked complex of CRL and ZIF-8 noted during the MD simulation. The possible H-bonding interactions that promote the catalysis of CRL have also been identified. We believe that this molecular modeling study will be a stepping stone for various future industrial applications of lipase, including biocatalysis, biofuel, biosensing, and the food industry.

## Figures and Tables

**Figure 1 biology-12-01051-f001:**
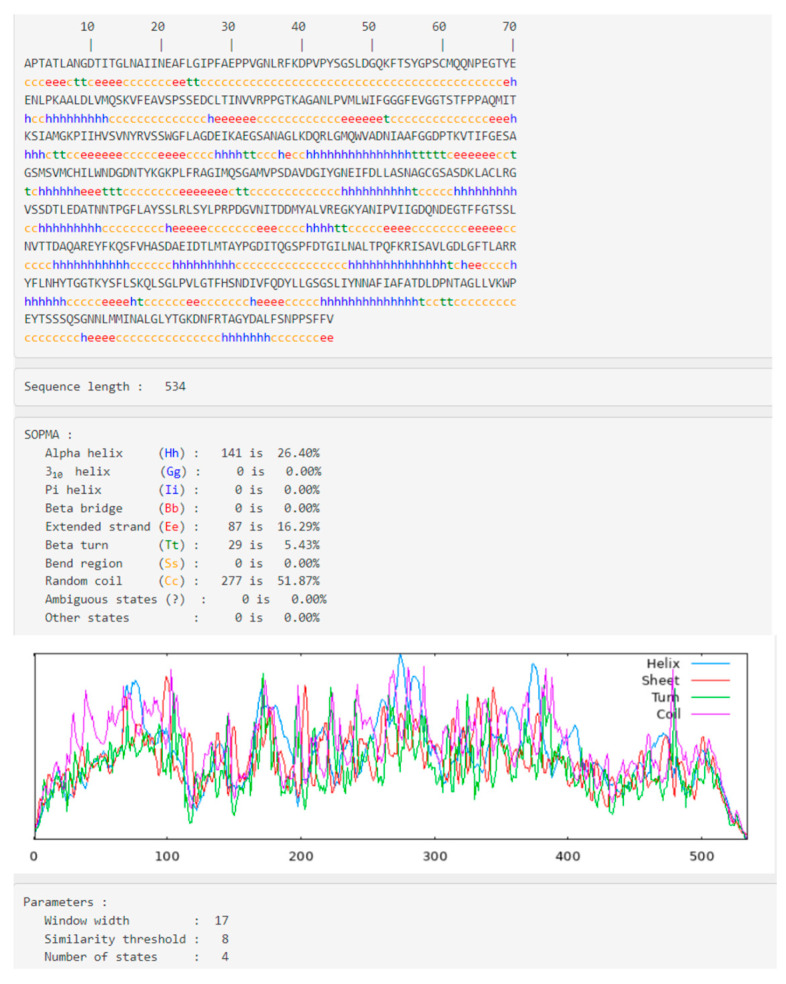
SOPMA analysis of the secondary structure of CRL, comprising helix, sheet, turn, loop, and coil. It has been found that CRL’s secondary structure is dominated by helices and random coils.

**Figure 2 biology-12-01051-f002:**
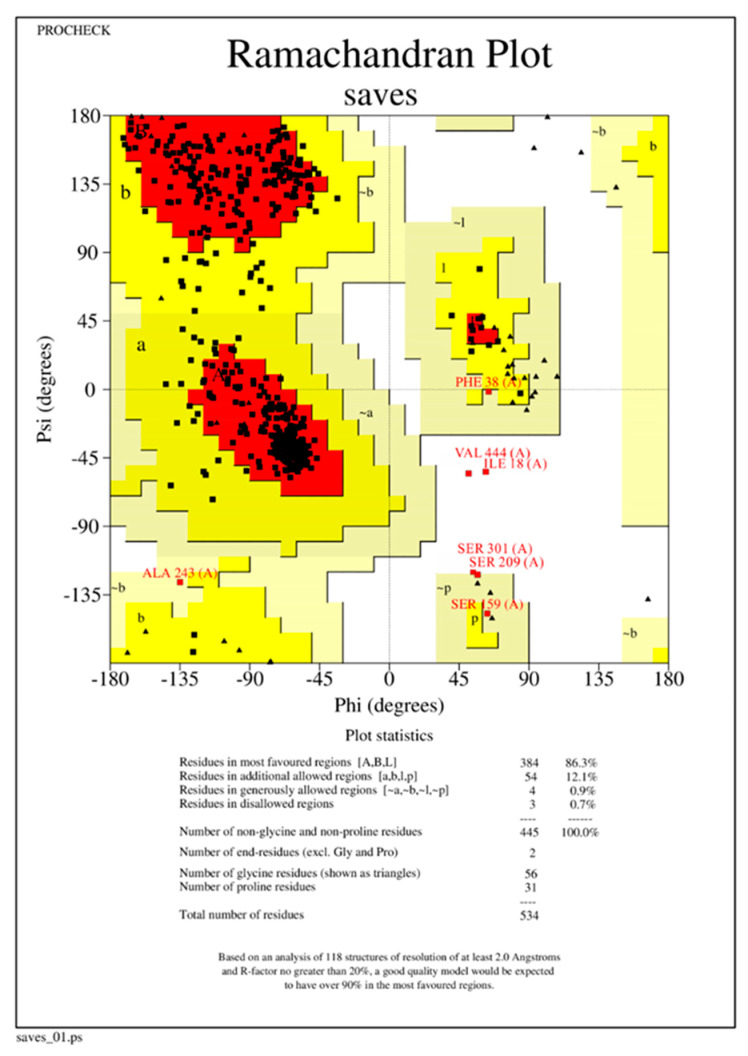
Ramachandran plot of a 3D model of CRL using PROCHECK displaying different regions of the modeled enzyme. The color coding used is as follows: red color shows favored regions, yellow color shows additional allowed regions, light yellow color displays generously allowed regions and, white color displays disallowed regions. The structural integrity of CRL is notably high, as the observation indicates that 99.30% of amino acid residues are situated within the favored regions, while only 0.70% of residues are present in the outer region.

**Figure 3 biology-12-01051-f003:**
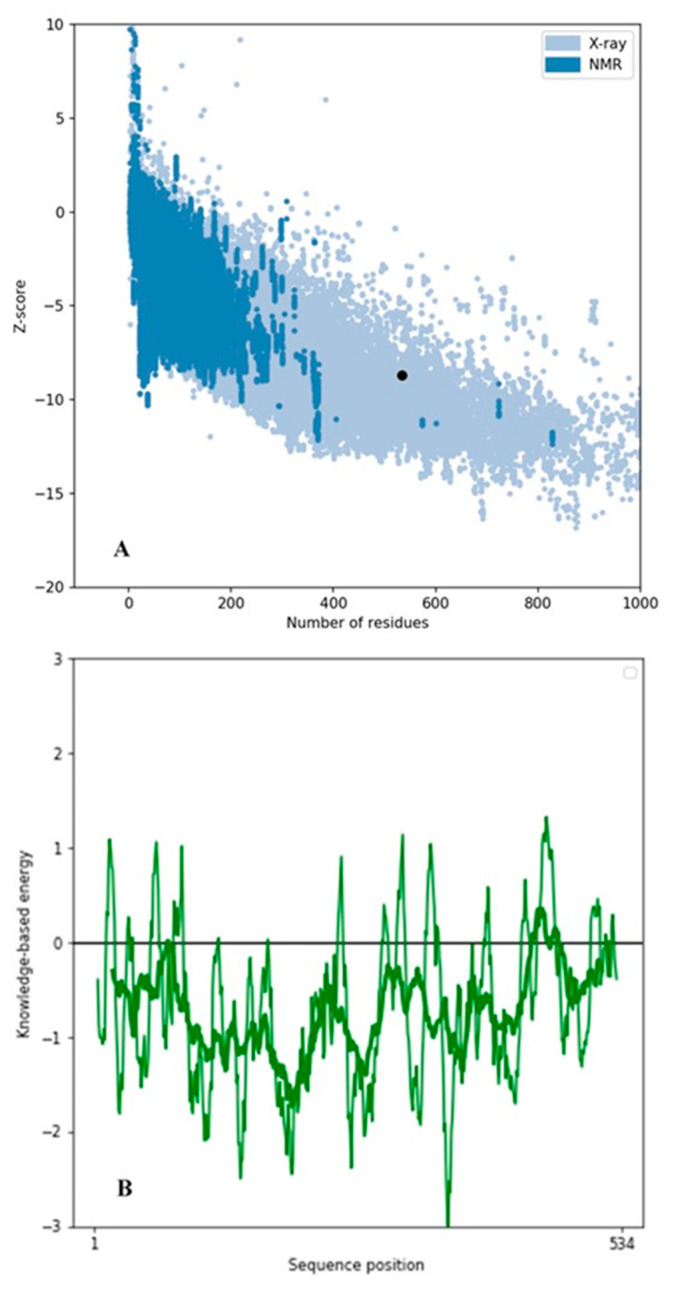
ProSA analysis of CRL. (**A**) The Z-score of CRL. The x-axis represents the number of residues, while the y-axis represents the Z-score. The black dot symbolizes the Z-score range of the CRL within the native conformation of crystal structures. The dark blue-colored region corresponds to the structure determined by Nuclear Magnetic Resonance (NMR), while the light blue-colored region corresponds to the structure determined by X-ray crystallography (XRC). (**B**) The energy plot for the predicted model of CRL. The X-axis represents the knowledge associated with energy, while the Y-axis represents the positional arrangement of amino acids within a sequence. The Z Score of −8.7 is within the acceptable range, indicating that the CRL model demonstrates a high level of quality. This is supported by the observation of a maximum number of residues displaying negative interaction energy.

**Figure 4 biology-12-01051-f004:**
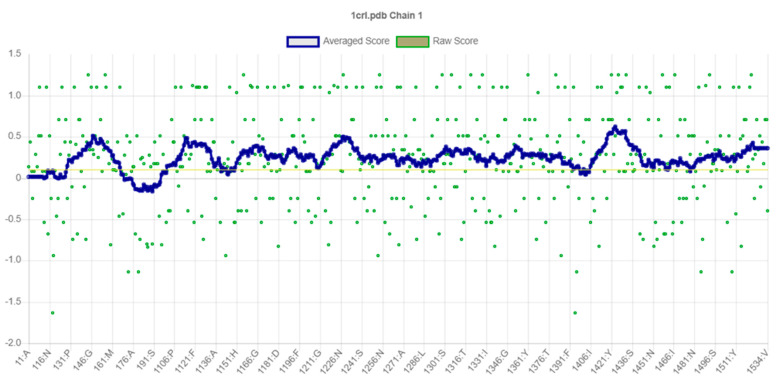
VERIFY-3D analysis of CRL. Approximately 86.33% of the residues exhibited a mean 3D-1D score ≥ 0.2, indicating that the proposed model demonstrated consistency with its frequency.

**Figure 5 biology-12-01051-f005:**
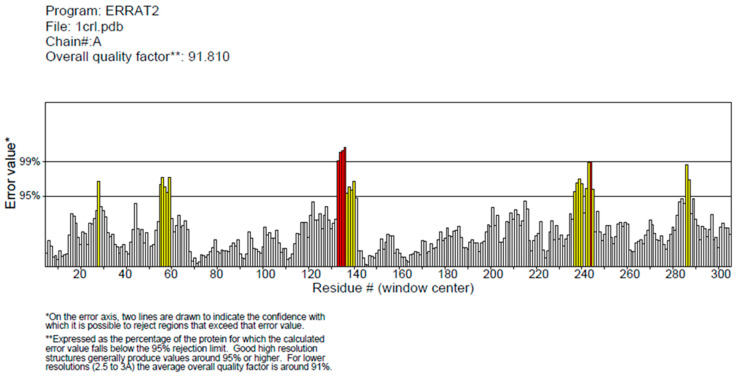
ERRAT analysis of CRL. The ERRAT plot is commonly utilized to assess the model and the amino acid environment. The error values of the model residues predicted by ERRAT are shown in the graph. CRL amino acid sequences are shown along the “x” axis, whereas error values are shown along the “y” axis. Error regions are depicted in red, misfolded regions in yellow, and regions with a lower error rate for protein folding are shown in white. The ERRAT plot analysis of the modeled structure resulted in an overall quality factor of 91.8095, indicating the significance of the model.

**Figure 6 biology-12-01051-f006:**
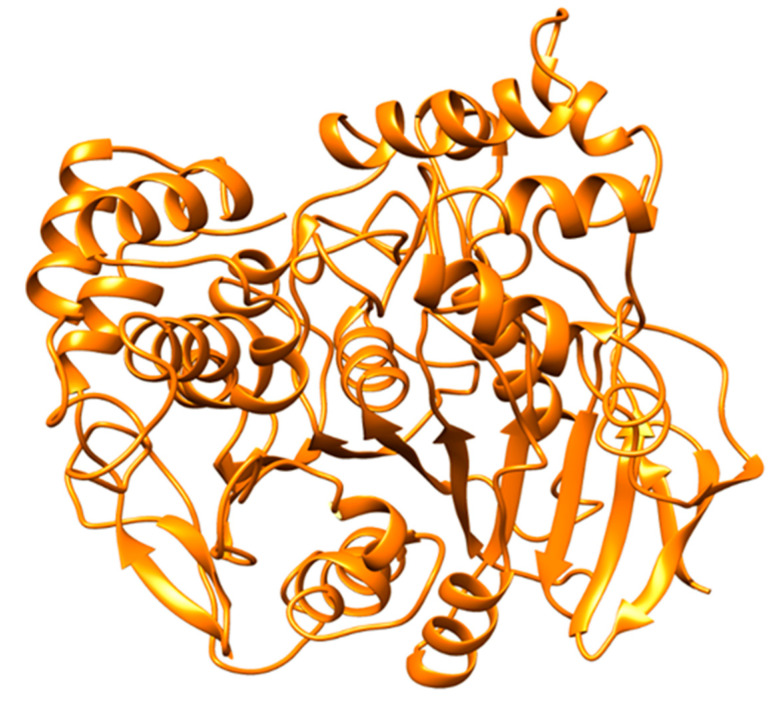
The crystal structure of the CRL enzyme (PDB ID: 1CRL). CRL is a member of the α/β hydrolase fold protein family and consists of a single domain.

**Figure 7 biology-12-01051-f007:**
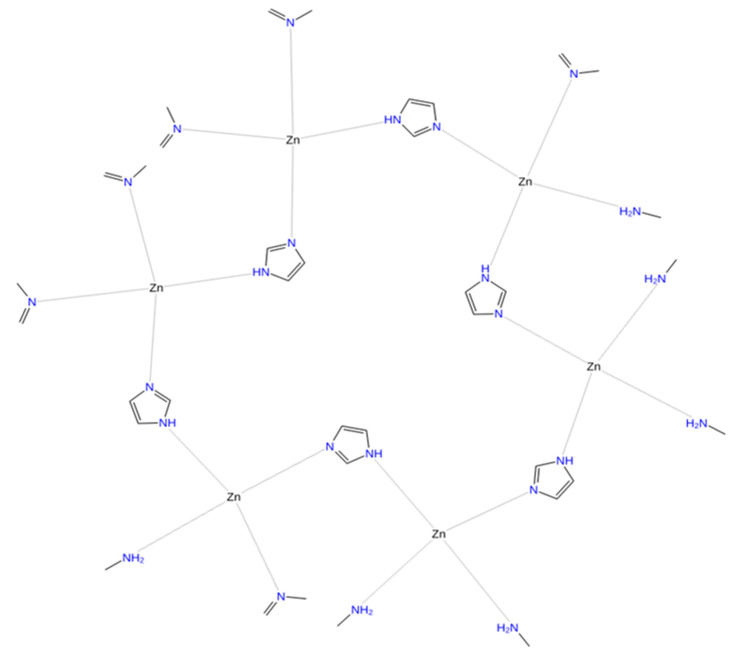
2D structure of the ligand ZIF-8 (CCDC deposit number 864309). ZIF-8 has a tetrahedral structure and comprises one Zn atom and four imidazolate linkers.

**Figure 8 biology-12-01051-f008:**
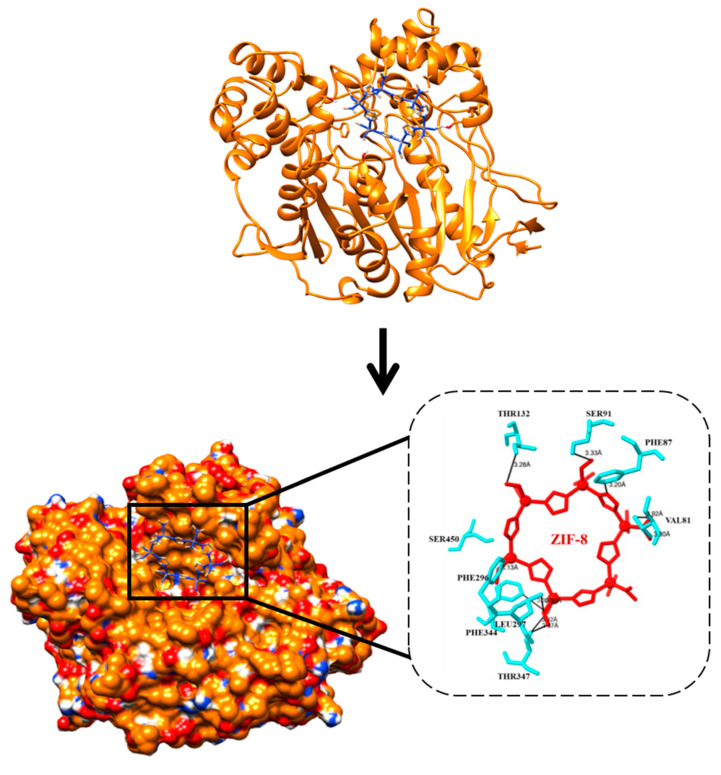
The docked complex of CRL and ligand ZIF-8. ZIF-8 has been shown to bind at the CRL’s interdimer interface, which is the protein’s largest binding pocket.

**Figure 9 biology-12-01051-f009:**
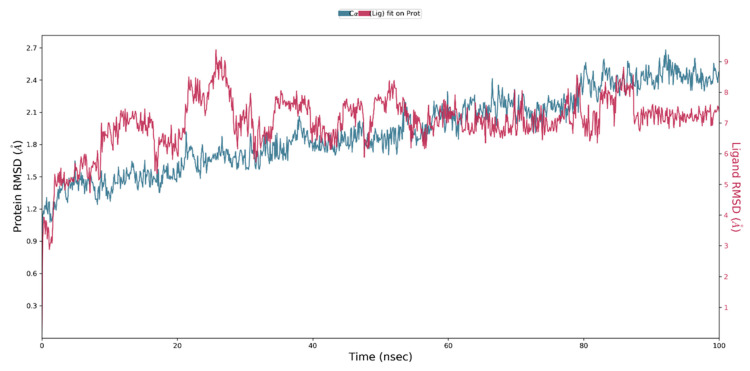
RMSD of CRL and ZIF-8 throughout the MD simulation. The RMSD plot showed that the ligand-protein complex was stable throughout the MD simulation.

**Figure 10 biology-12-01051-f010:**
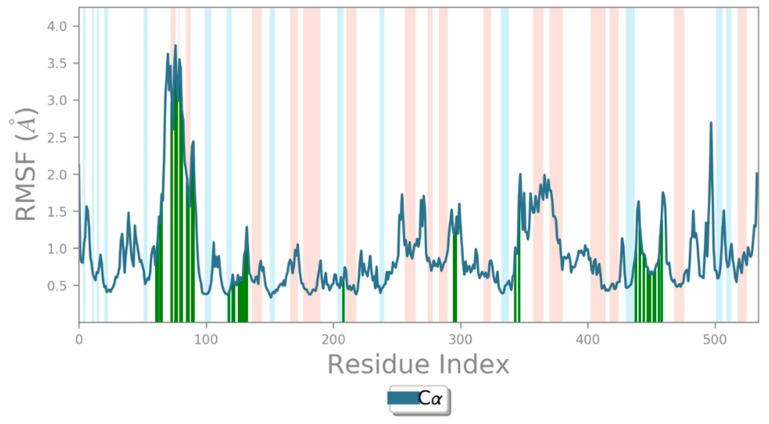
RMSF of CRL and ZIF-8 throughout the MD simulation. The RMSF graph confirms the protein structure is stable, with fewer variations.

**Figure 11 biology-12-01051-f011:**
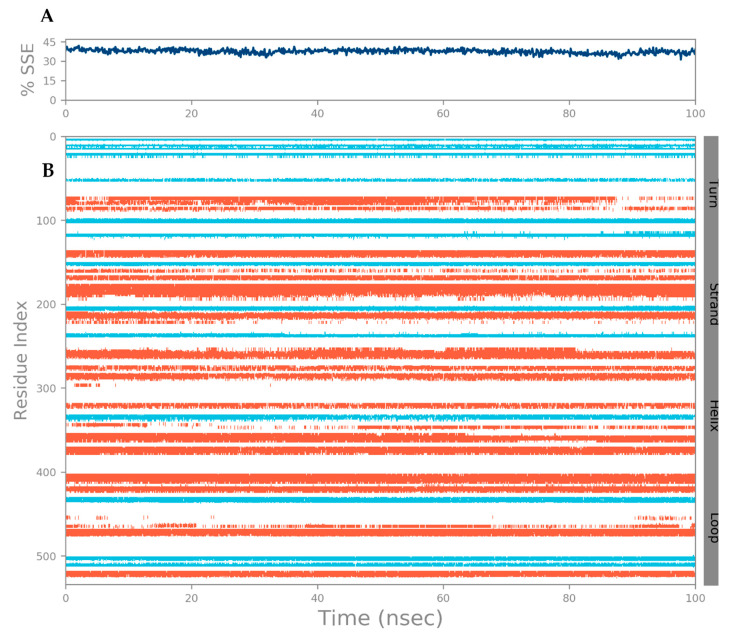
(**A**) The SSE distribution by residue index throughout the protein structure; (**B**) SSE composition for each trajectory frame over the course of the simulation; and the plot at the bottom monitors each residue and its SSE assignment over time in ns.

**Figure 12 biology-12-01051-f012:**
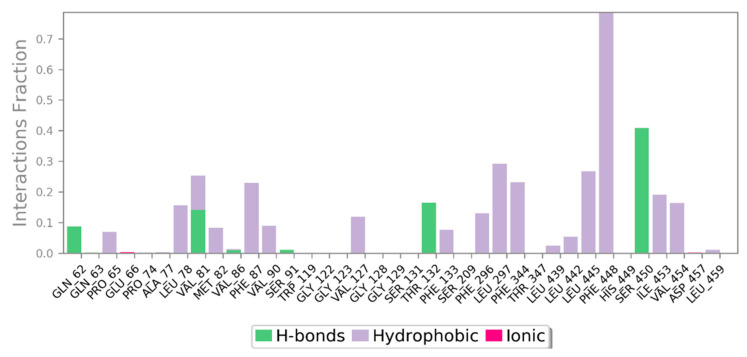
The bonding interactions between the CRL and the ZIF-8 were analyzed during the MD simulation. These interactions can be categorized into various types, including hydrogen bonds (H-bonds), hydrophobic interactions, ionic contacts, and water bridges. These different types of interactions are commonly observed in protein-ligand systems.

**Figure 13 biology-12-01051-f013:**
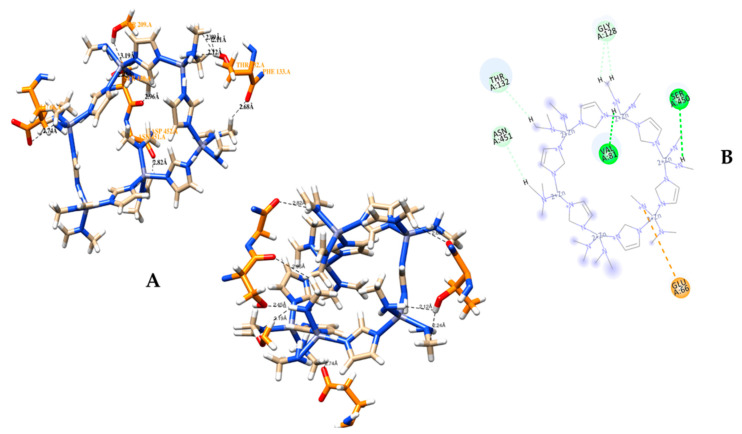
Hydrogen bonding interactions between CRL and ZIF-8. (**A**) The 3D and (**B**) 2D representations. The significant hydrogen bonding interactions established by CRL and ZIF-8 involving the amino acid residues Glu-66, Thr-132, Ser-209, Ser-450, and Asn-451 suggest potential hydrogen bonding sites within CRL. The residues Thr-132, Ser-209, Ser-450, and Asn-451 exhibit hydrogen bonding interactions with the backbone of ZIF-8.

**Figure 14 biology-12-01051-f014:**
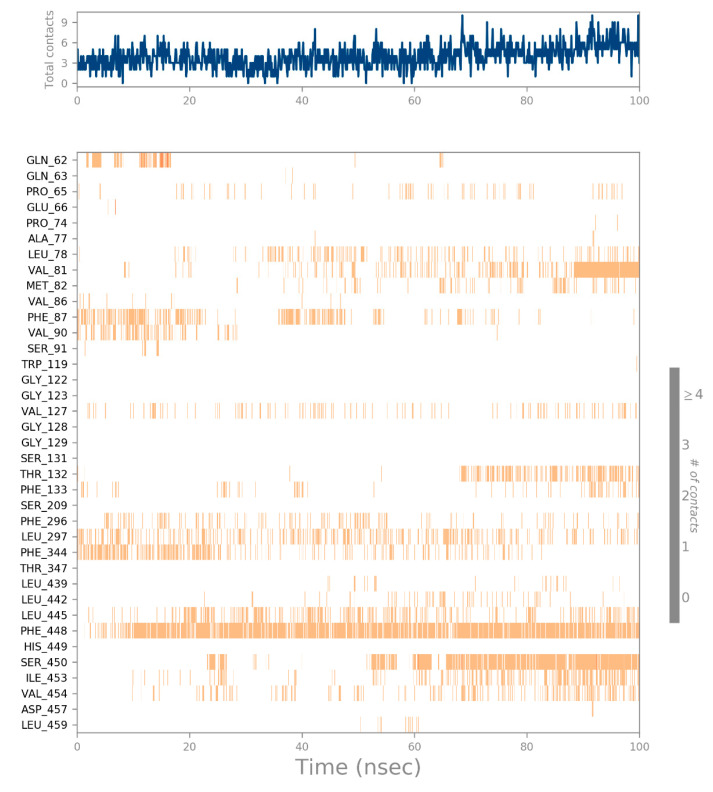
The residues of CRL that interact with the ZIF-8 in each trajectory frame. Multiple ligand interactions are made by some residues, as seen by the scale on the right side of the plot.

**Figure 15 biology-12-01051-f015:**
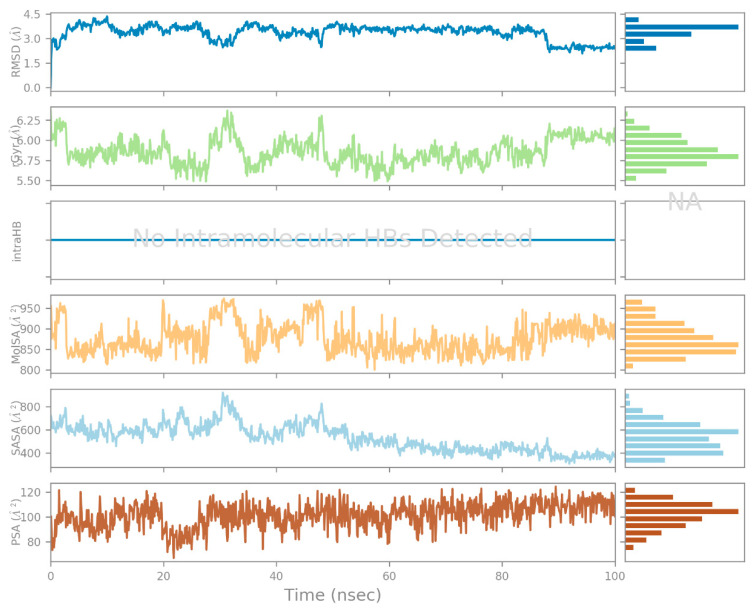
RMSD, rGyr, MolSA, SASA, and PSA of the CRL-ZIF-8 complex throughout the MD simulation. These interactions demonstrated that the docked complex of receptor and ligand is stable over the course of the MD simulation, with all existing contacts being maintained.

**Table 1 biology-12-01051-t001:** Hydrogen bonding interaction between CRL and ZIF-8 before MD.

Sr. No.	Possible Hydrogen Bonding Interactions between Active Site Residues of CRL and ZIF-8 before MD	Distance in Å
1	THR 347 HG1 ------------ ZIF 1 C:	2.87
2	PHE 296 CE1 ------------ ZIF 1 N:	3.13
3	ZIF 1 C ------------ PHE 87 CE1:	3.20
4	PHE 344 CE1 ------------ ZIF 1 C:	3.30
5	LEU 297 CD1 ------------ ZIF 1 N:	3.30
6	ZIF 1 C ------------ VAL 81 CG2:	3.30
7	ZIF 1 C ------------ THR 347 HG1:	3.32
8	ZIF 1 C ------------ SER 91 HG:	3.33
9	ZIF 1 C ------------ VAL 81 CG1:	3.92

**Table 2 biology-12-01051-t002:** Hydrogen bonding interaction between CRL and ZIF-8 after MD.

Sr. No.	Hydrogen Bond Interactions between Amino Acid Residues of CRL with ZIF-8 after MD	Distance in Å
1	ZIF.het H__2------------THR 132.A HG1	2.12
2	ZIF.het H__1------------SER 209.A HG	3.19
3	ZIF.het H__1------------SER 450.A HG	2.45
4	ZIF.het 1HXT------------GLU 66.A OE2	2.74
5	ZIF.het 3HXT------------THR 132.A O	2.68
6	ZIF.het 2HXT------------SER 450.A O	2.96
7	ZIF.het 1HXT------------ASN 451.A O	2.82
8	ZIF.het 2HXT------------THR 132.A HG1	2.11
9	ZIF.het 3HXT------------THR 132.An HG1	2.49

## Data Availability

The experimental data are available upon request.

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
