# Peer review of "Molecular Modeling Insights into Metal-Organic Frameworks (MOFs) as a Potential Matrix for Immobilization of Lipase: An In Silico Study"

_biology, 2023, doi:10.3390/biology12081051_

Round 1
Reviewer 1 Report
This work by Patil et al focuses on studying the interaction between Candida rugosa lipase (CRL) and the Zeolitic imidazolate framework (ZIF-8). The researchers employ molecular modeling techniques, including molecular docking and molecular dynamics simulation, to comprehensively investigate this interaction. The goal is to enhance the understanding of CRL-MOF interactions and explore their potential practical applications in biocatalysis. This work contributes to the existing literature by specifically examining the CRL-ZIF-8 interaction, which has been less explored compared to the interaction of lipase with other ligands. The manuscript is well-structured, and the writing is clear and concise, allowing for easy comprehension of the research presented. The methods section provides sufficient details regarding the computational techniques used, enabling the reproducibility of the study. I recommend this manuscript for publication, pending the authors' attention to the minor revisions mentioned above.
1. Page 1 line 80. Check the sentence MOFs are highly organized microporous crystalline…” The microporous is not accurate, please check. MOF could be micro-/meso- porous.
2. Page 1 line 83. “…metal ions, and amines, nitrates, carboxylates, sulfonates, and phosphates are regularly used as organic linkers…” The MOFs typically are carboxylate and nitrogen-containing heterocyclic components coordinated.
3. There are many missing references on enzyme-MOF. E.g. (ACS Cent. Sci. 2020, 6, 9, 1497–1506.) It highlights the importance of investigating enzyme-MOF interactions/enzyme structure changes in MOFs. (Chem. Rev. 2021, 121, 1077−1129.) It comprehensively summarized current works from different aspects.
4. For Figure 13, it is hard to distinguish the hydrogen bonding sites, please provide figures with higher resolution. Also, use different models/colors to represent the frameworks of ZIF-8 and amino acids of enzyme for easier reading.
5. The ZIF-8 is a microporous MOF with a pore size of ~ 1nm. The experimental in-situ encapsulation was conducted by crystallization, see reference (Nat. Commu. 2019, 10, 5165). The enzyme must be dramatically restrained by the framework. How to address this issue in simulation and experiments? It would be beneficial if the authors could provide additional information regarding the enzyme location/arrangement inside MOF for the simulation.
6. What are the difficulties in the MD modeling enzymes-MOFs? Why there are very limiting works developed? An extensive discussion is suggested.
Author Response
The authors are grateful to the reviewer and editor for their valuable input to the manuscript (MS). The authors have carefully revised the MS as per the suggestions. To facilitate your review of our revisions, the following is a point-by-point response to the questions and comments delivered. We have color-coded revised MS as text. The changes that have been made as concerns raised by reviewers are marked red in the revised MS. The responses to the concerns raised by reviewers are below and are color-coded as follows: a) Comments from editors or reviewers are shown as black text; b) Our responses are shown as red text. We believe that the contents and the clarity of our paper are much improved in the revised version and hope that these revisions persuade you to accept our submission.
Response to reviewer's comments
Reviewer #1:
This work by Patil et al focuses on studying the interaction between Candida rugosa lipase (CRL) and the Zeolitic imidazolate framework (ZIF-8). The researchers employ molecular modeling techniques, including molecular docking and molecular dynamics simulation, to comprehensively investigate this interaction. The goal is to enhance the understanding of CRL-MOF interactions and explore their potential practical applications in biocatalysis. This work contributes to the existing literature by specifically examining the CRL-ZIF-8 interaction, which has been less explored compared to the interaction of lipase with other ligands. The manuscript is well-structured, and the writing is clear and concise, allowing for easy comprehension of the research presented. The methods section provides sufficient details regarding the computational techniques used, enabling the reproducibility of the study. I recommend this manuscript for publication, pending the authors' attention to the minor revisions mentioned above.
Response: The authors sincerely appreciate the positive feedback and insightful comments and suggestions from the reviewer for revising the MS and embracing it wholeheartedly. The authors believe that the reviewer's issues are generally straightforward to address satisfactorily, and we have substantially revised the MS to this effect. As suggested by the reviewer, the authors have reviewed the entire MS carefully and would like to highlight that the revised MS is now well-furnished and improved.
- Page 1 line 80. Check the sentence MOFs are highly organized microporous crystalline…" The microporous is not accurate, please check. MOF could be micro-/meso- porous.
Response: The authors apologize profusely for the oversight. It has been modified (See Line no. 80).
- Page 1 line 83. "…metal ions, and amines, nitrates, carboxylates, sulfonates, and phosphates are regularly used as organic linkers…" The MOFs typically are carboxylate and nitrogen-containing heterocyclic components coordinated.
Response: The authors agree with the reviewer's comment. However, some MOFs have sulfonates and phosphates used as organic linkers [1-3].
- There are many missing references on enzyme-MOF. E.g. (ACS Cent. Sci. 2020, 6, 9, 1497–1506.) It highlights the importance of investigating enzyme-MOF interactions/enzyme structure changes in MOFs. (Chem. Rev. 2021, 121, 1077−1129.) It comprehensively summarized current works from different aspects.
Response: Per the suggestion of the reviewer, the given references have been added to the revised MS (See Line nos. 81, 153).
- For Figure 13, it is hard to distinguish the hydrogen bonding sites; please provide figures with higher resolution. Also, use different models/colors to represent the frameworks of ZIF-8 and amino acids of enzyme for easier reading.
Response: The quality of Figure 13 has been improved, and the higher resolution figure has been updated in the revised MS. Furthermore, the authors would like to highlight that these figures have been automatically generated through the software/online programs. Thus, authors are unable to edit/change these figures.
- The ZIF-8 is a microporous MOF with a pore size of ~ 1nm. The experimental in-situ encapsulation was conducted by crystallization, see reference (Nat. Commu. 2019, 10, 5165). The enzyme must be dramatically restrained by the framework. How to address this issue in simulation and experiments? It would be beneficial if the authors could provide additional information regarding the enzyme location/arrangement inside MOF for the simulation.
Response: The information has been included in section 4, "Discussion," in the revised MS (See Line nos. 365-448).
- What are the difficulties in the MD modeling enzymes-MOFs? Why there are very limiting works developed? An extensive discussion is suggested.
Response: An extensive discussion regarding this issue has been represented in the revised MS in section 4, "Discussion" (See Line nos. 449-530).
References
- Liang, S.; Wu, X.L.; Xiong, J.; Zong, M.H.; Lou, W.Y. Metal-Organic Frameworks as Novel Matrices for Efficient Enzyme Immobilization: An Update Review. Coord Chem Rev 2020, 406.
- Mehta, J.; Bhardwaj, N.; Bhardwaj; S.K., Kim, K.H.; Deep, A. Recent Advances in Enzyme Immobilization Techniques: Metal-organic Frameworks as Novel Substrates. Coord Chem Rev 2016, 322, 30-40.
- Shimizu, G.K.; Vaidhyanathan, R.; Taylor, J.M. Phosphonate and Sulfonate Metal Organic Frameworks. Chem Soc Rev 2009, 38(5), 1430-1449.

Reviewer 2 Report
Molecular docking and molecular dynamics methods were used for simulation of lipase immobilization of MOFs. The subject is potentially interesting. However, there are some points which should be addressed and clarified in the revised version, as mentioned below:
1. Could the authors comment on the role of surface charge of the agents (including MOFs) in the lipase immobilization?
2. It has been mentioned that “Lipase is the most frequently acknowledged enzyme in diverse fields of industrial biotechnology and microbiology, and it is currently regarded as one of the key participants in numerous industrial processes”. This statement can be further supported by (Ugi four-component assembly process: an efficient approach for one-pot multifunctionalization of nanographene oxide in water and its application in lipase immobilization), too.
3. Could the authors comment on the hydrophilicity/hydrophobicity of the MOFs? Then, what is the role of surface roughness of the MOFs in the hydrophilicity/hydrophobicity?
4. In Figure 2 there are some overlapping for the labels. Please clarify. There is also some overlapping in Figure 9.
5. It has been mentioned that “… MOFs have a number of outstanding characteristics, such as …”. This can be further completed by considering heat energy storage [Metal-organic frameworks (MOF) based heat transfer: A comprehensive review], too.
6. The quality of Figure 3 does not seem suitable for publication. It should be improved.
7. It has been mentioned that “As a result, they have received a lot of attention for scientific studies and real-world applications in various domains, embracing chemical catalysis, biosensing and detection, gas adsorption and separation, and drug loading and delivery.”. This needs to be supported by e.g., (Green metal-organic frameworks (MOFs) for biomedical applications) and (Metal–organic framework (MOF)-based drug/cargo delivery and cancer therapy) as some recent reviews in this regard.
8. In Some figures such as Figure 4, it seems that the gray color was used for drawing the axes and labels. The use of black color is recommended for better clarity.
9. It has been stated that “… tools like molecular docking and molecular dynamics (MD) simulation can make predictions about enantioselectivity, potential catalytic deactivations, or compound affinity, which arise from interactions between an enzyme, substrate, and solvent, more rapidly and with less need for expensive materials”. This can be further supported by some new works such as (SOEing PCR/docking optimization of protein A-G/scFv-Fc-bioconjugated Au nanoparticles for interaction with meningitidis bacterial antigen).
10. There are some descriptions within Figure 5. These statements should be transferred to the caption of the figure or the main text. At the present case the figure captions are not so informative.
11. Can be assigned any dispersion for the data points presented in Figure 12?
Some technical revisions are required.
Author Response
The authors are grateful to the reviewer and editor for their valuable input to the manuscript (MS). The authors have carefully revised the MS as per the suggestions. To facilitate your review of our revisions, the following is a point-by-point response to the questions and comments delivered. We have color-coded revised MS as text. The changes that have been made as concerns raised by reviewers are marked red in the revised MS. The responses to the concerns raised by reviewers are below and are color-coded as follows: a) Comments from editors or reviewers are shown as black text; b) Our responses are shown as red text. We believe that the contents and the clarity of our paper are much improved in the revised version and hope that these revisions persuade you to accept our submission.
Response to reviewer's comments
Reviewer #2:
Molecular docking and molecular dynamics methods were used for simulation of lipase immobilization of MOFs. The subject is potentially interesting. However, there are some points which should be addressed and clarified in the revised version, as mentioned below:
Response: The authors sincerely appreciate the reviewer's positive feedback and insightful comments and suggestions for revising the MS and embracing it wholeheartedly. The authors believe the reviewer's issues are generally straightforward to address satisfactorily, and we have substantially revised the MS to this effect. As suggested by the reviewer, the authors have reviewed the entire MS carefully and would like to highlight that the revised MS is now well-furnished and improved.
- Could the authors comment on the role of surface charge of the agents (including MOFs) in the lipase immobilization?
Response: The surface charge of agents, including MOFs, is crucial in immobilizing lipase enzymes. The surface charge affects the interaction between the agent and the lipase, influencing the immobilized lipase's adsorption, stability, and catalytic activity. The following are the vital aspects related to the role of surface charge in lipase immobilization:
1) Electrostatic interactions: Lipases often carry a net charge due to charged amino acid residues on their surface. The surface charge of lipases can interact with the surface charge of the immobilization agent, such as a MOF. If the immobilization agent has a charge opposite to that of the lipase, electrostatic attraction can facilitate lipase binding to the MOF surface.
2) Surface adsorption: The immobilization agent's surface charge can influence lipase adsorption. Electrostatic interactions between the charged regions of the lipase and the MOF surface can promote adsorption by providing favorable binding sites. The surface charge of the MOF can affect the adsorption capacity and stability of the immobilized lipase.
3) Orientation and conformation: The immobilization agent's surface charge can influence the immobilized lipase's orientation and conformation. Electrostatic interactions between the lipase and the MOF surface can drive specific orientations of the lipase, resulting in the preferred positioning of the active site for catalytic activity. The surface charge can also impact the conformational stability of the immobilized lipase.
4) Stability and leakage: The surface charge of the immobilization agent, particularly the MOF, can affect the stability of the immobilized lipase. Electrostatic interactions can provide additional stabilization to the lipase structure, preventing denaturation or loss of activity. Moreover, the surface charge can impact the leakage of the immobilized lipase from the MOF, ensuring its retention and improved operational stability.
5) Substrate accessibility: The immobilization agent's surface charge can influence the accessibility of substrates to the active site of the immobilized lipase. Electrostatic interactions between the charged residues of the lipase and the MOF surface can affect the diffusion of substrates, potentially enhancing the catalytic efficiency of the immobilized lipase.
- It has been mentioned that "Lipase is the most frequently acknowledged enzyme in diverse fields of industrial biotechnology and microbiology, and it is currently regarded as one of the key participants in numerous industrial processes". This statement can be further supported by (Ugi four-component assembly process: an efficient approach for one-pot multifunctionalization of nanographene oxide in water and its application in lipase immobilization), too.
Response: The given reference supporting the above statements has been added to the revised MS per the reviewer's suggestion " (See Line no. 62).
- Could the authors comment on the hydrophilicity/hydrophobicity of the MOFs? Then, what is the role of surface roughness of the MOFs in the hydrophilicity/hydrophobicity?
Response: The property of hydrophilicity or hydrophobicity in MOFs holds significant importance as it can greatly impact their behavior and potential applications. The term "hydrophilic" describes the property of the MOF surface to attract or have an affinity for water, while "hydrophobic" refers to the property of the MOF surface to repel or have an aversion to water. Various factors, such as the selection of metal ions, organic ligands, and functionalization techniques, can influence the hydrophilicity/hydrophobicity of MOFs.
1) Hydrophilic MOFs: Hydrophilic MOFs possess surfaces that demonstrate a distinct attraction towards water molecules. The phenomenon of hydrophilicity can be attributed to the existence of hydrophilic functional groups, such as hydroxyl (-OH), carboxyl (-COOH), or amine (-NH2) groups, either on the ligands or within the structure MOFs. They exhibit a pronounced affinity for water molecules, facilitating their efficient absorption and retention, thereby leading to substantial water uptake capabilities. Also, they display favorable characteristics that render them appropriate for various applications, including but not limited to water adsorption, separation, and catalysis within aqueous environments.
2) Hydrophobic MOFs: Hydrophobic MOFs possess surfaces demonstrating a strong aversion to water molecules, resulting in a low affinity for water. Hydrophobicity can be attained by utilizing hydrophobic organic ligands or integrating nonpolar moieties, such as alkyl chains, within the structure of MOFs. They generally demonstrate reduced solubility in aqueous environments and display diminished water absorption capacities compared to hydrophilic MOFs. These MOFs are frequently employed in various applications such as gas adsorption, separation of hydrophobic molecules, and storage.
The hydrophilicity/hydrophobicity of MOFs may exhibit variability depending upon the particular ligands, metal ions, and surface modifications employed. Various experimental characterization techniques, such as contact angle measurements, water adsorption isotherms, and solubility studies, can be employed to assess and compare MOFs' hydrophilic and hydrophobic properties quantitatively. The hydrophilic or hydrophobic nature of MOFs can be modified using a range of different approaches.
The term "surface roughness" in the context of MOFs pertains to the non-uniformities, deviations, or characteristics observed on the external surface of the MOFs. The surface roughness can influence the hydrophilicity or hydrophobicity of MOFs. Surface roughness influences the hydrophilicity or hydrophobicity of MOFs through two primary mechanisms:
- a) The contact angle: Variations can influence the contact angle between water droplets and the surface of a MOF in surface roughness. Typically, surfaces with greater roughness exhibit increased contact angles, thereby rendering them more hydrophobic.
- b) Surface area: Rough surfaces exhibit a greater surface area in comparison to smooth surfaces. The augmented surface area facilitates a greater number of potential binding sites for water molecules, thereby enhancing the probability of hydrophilic interactions. Consequently, an increase in surface roughness can potentially augment the hydrophilic properties of MOFs by facilitating a greater degree of interaction between water molecules and the surface.
It is crucial to acknowledge that the precise impacts of surface roughness on the hydrophilic or hydrophobic properties can vary based on several factors, such as the characteristics of the MOFs, the scale of roughness, and the interactions between the surface of the MOF and water molecules. Moreover, MOFs' hydrophilic or hydrophobic behavior can be influenced by the presence of other functional groups or modifications on their surface.
- In Figure 2 there are some overlapping for the labels. Please clarify. There is also some overlapping in Figure 9.
Response: The authors apologize profusely for the oversight. Figures 2 and 9 have been corrected now in the revised MS.
- It has been mentioned that "… MOFs have a number of outstanding characteristics, such as …". This can be further completed by considering heat energy storage [Metal-organic frameworks (MOF) based heat transfer: A comprehensive review], too.
Response: It has been modified (See Line no. 90).
- The quality of Figure 3 does not seem suitable for publication. It should be improved.
Response: The authors apologize profusely for the oversight. The quality of Figure 3 has been improved now in the revised MS.
- It has been mentioned that "As a result, they have received a lot of attention for scientific studies and real-world applications in various domains, embracing chemical catalysis, biosensing and detection, gas adsorption and separation, and drug loading and delivery.". This needs to be supported by e.g., (Green metal-organic frameworks (MOFs) for biomedical applications) and (Metal–organic framework (MOF)-based drug/cargo delivery and cancer therapy) as some recent reviews in this regard.
Response: The given references supporting the above statements have been added along with some new references to the revised MS per the reviewer's suggestion (See Line no. 96).
- In Some figures such as Figure 4, it seems that the gray color was used for drawing the axes and labels. The use of black color is recommended for better clarity.
Response: The authors appreciate the suggestion; however, the authors would like to shed light on the fact that these figures have been automatically generated through the software/online programs. Thus, authors are unable to edit/change these figures.
- It has been stated that "… tools like molecular docking and molecular dynamics (MD) simulation can make predictions about enantioselectivity, potential catalytic deactivations, or compound affinity, which arise from interactions between an enzyme, substrate, and solvent, more rapidly and with less need for expensive materials". This can be further supported by some new works such as (SOEing PCR/docking optimization of protein A-G/scFv-Fc-bioconjugated Au nanoparticles for interaction with meningitidis bacterial antigen).
Response: The given reference supporting the above statements has been added to the revised MS per the reviewer's suggestion (See Line no. 108).
- There are some descriptions within Figure 5. These statements should be transferred to the caption of the figure or the main text. At the present case the figure captions are not so informative.
Response: The authors appreciate the suggestion. As a reflection of the reviewer's comment, the changes have been made. Moreover, figure captions of each figure have been improved and expanded in the revised MS.
- Can be assigned any dispersion for the data points presented in Figure 12?
Response: The bonding interactions between the CRL and the ZIF-8 were found as hydrogen bonds (H-bonds), hydrophobic interactions, and ionic contacts during MD simulation, depending upon the respective groups interacting. We have presented bonding interactions based on the structures obtained after the MD simulation. Subsequently, there's no dispersion available.
Comments on the Quality of English Language
Some technical revisions are required.
Response: As suggested by the reviewer, the authors have reviewed the entire MS carefully and would like to highlight that the revised MS is now well-furnished and improved and is now technically sound.

Round 2
Reviewer 2 Report
The manuscript has been revised based on the comments and so can be considered for publication.